# The Multifaceted Role of Glutathione S-Transferases in Health and Disease

**DOI:** 10.3390/biom13040688

**Published:** 2023-04-18

**Authors:** Aslam M. A. Mazari, Leilei Zhang, Zhi-Wei Ye, Jie Zhang, Kenneth D. Tew, Danyelle M. Townsend

**Affiliations:** 1Department of Cell and Molecular Pharmacology and Experimental Therapeutics, Medical University of South Carolina, 70 President Street, DDB410, Charleston, SC 29425, USA; 2Department of Pharmaceutical and Biomedical Sciences, Medical University of South Carolina, 274 Calhoun Street, MSC141, Charleston, SC 29425, USA

**Keywords:** Glutathione S-transferases (GSTs), antioxidants, cancer-cell signaling, cell survival, chemoresistance, xenobiotic compounds, metabolism, GST inhibitors, glutathionylation, oxidative stress, JNK, apoptosis, GST Polymorphism, SARS-CoV-2, COVID-19

## Abstract

In humans, the cytosolic glutathione S-transferase (GST) family of proteins is encoded by 16 genes presented in seven different classes. GSTs exhibit remarkable structural similarity with some overlapping functionalities. As a primary function, GSTs play a putative role in Phase II metabolism by protecting living cells against a wide variety of toxic molecules by conjugating them with the tripeptide glutathione. This conjugation reaction is extended to forming redox sensitive post-translational modifications on proteins: S-glutathionylation. Apart from these catalytic functions, specific GSTs are involved in the regulation of stress-induced signaling pathways that govern cell proliferation and apoptosis. Recently, studies on the effects of GST genetic polymorphisms on COVID-19 disease development revealed that the individuals with higher numbers of risk-associated genotypes showed higher risk of COVID-19 prevalence and severity. Furthermore, overexpression of GSTs in many tumors is frequently associated with drug resistance phenotypes. These functional properties make these proteins promising targets for therapeutics, and a number of GST inhibitors have progressed in clinical trials for the treatment of cancer and other diseases.

## 1. Introduction

Glutathione transferases (GSTs), also referred to as glutathione *S*-transferases, belong to the supergene family of phase II detoxification enzymes that are ubiquitously present in almost all cellular life forms. On the basis of their sub-cellular distribution, GSTs are classified into three major protein families as either Cytosolic, Mitochondrial or Kappa and Microsomal (also known as Membrane-Associated Proteins in Eicosanoid and Glutathione (MAPEG)) [1]. The cytosolic GSTs constitute the largest family and are sub-divided into seven distinct classes based on their amino acid sequence and other structural similarities, represented by Greek letter names with alphanumeric letter designations, namely alpha (A), mu (M), omega (O), pi (P), sigma (S), theta (T), and zeta (Z) [2,3,4]. Cytosolic GSTs are structurally distinct from mitochondrial and microsomal classes of enzymes and are composed of two subunits (~25 kDa each), either homodimers of a single gene product or heterodimers encoded by a different gene. Each subunit of the dimeric isozyme consists of two functional domains, the more conserved N-terminal domain containing catalytically active cysteine, serine, or tyrosine residues, and the C-terminal domain. There are two substrate binding sites in each subunit: the GSH binding site or G-site and an adjacent H-site for binding structurally diverse hydrophobic xenobiotics or the products of oxidative stress [1,5]. As principal phase II detoxification enzymes, GSTs protect living cells by catalyzing the conjugation of glutathione (GSH) to a wide variety of electrophilic molecules of both endogenous and exogenous origin. GSH is a tripeptide (γ-l-glutamyl-l-cysteinyl glycine) synthesized in the cytoplasm of every cell in a two-step ATP-requiring enzymatic process catalyzed by glutamate-cysteine ligase and glutathione synthetase enzymes. GST tissue distribution and expression levels vary according to the class. A recent review summarizes the tissue distribution of soluble GSTs [6]. The variations in GST tissue distribution suggest possible differences in the ways by which individual human tissues can detoxify or otherwise handle xenobiotics and or drugs. Furthermore, certain chemicals, including those occurring naturally in fruits and cruciferous vegetables can act as inducers of GST genes through a range of responsive elements and such inductions are part of GSTs adaptive response mechanisms to chemical insult caused by electrophiles [7].

Apart from their essential role as detoxification enzymes, GSTs are involved in several other important functions such as cell signaling, post-translational modifications, and chemotherapeutic drug resistance [8]. For example, the pi and mu classes of GSTs modulate the mitogen-activated protein kinase (MAPK) signaling pathway responsible for stress response, cell proliferation, and apoptosis via direct interactions with c-Jun N-terminal kinase 1 (JNK1) and apoptosis signal-regulating kinase (ASK1) [8,9]. Furthermore, GSTs are known to facilitate protein S-glutathionylation reactions, and a number of proteins have been shown to be common substrates for GST-mediated protein S-glutathionylation including protein disulfide isomerase (PDI), p53, and peroxiredoxin-VI (Prdx-VI) [10]. Overexpression of GSTs, particularly GSTP1-1 is often considered as a possible mechanism of tumor cell drug resistance [11,12,13]. Hence, GSTs remain a viable therapeutic target, and inhibitors of GST catalytic activity have emerged as potential therapeutic tools in cancer cell drug resistance [14,15]. This review focuses on the diverse role these versatile enzymes play in human health and disease.

## 2. Role in Detoxification

GSTs function in cellular protection by catalyzing the conjugation of GSH with numerous hydrophobic and electrophilic intermediates, including many carcinogens, therapeutic agents, and products of oxidative stress, rendering them less toxic and facilitating their export from the cell. An overview of xenobiotic detoxification is shown below in (Figure 1).

The human genome encodes 16 cytosolic GST enzymes with overlapping, but non-identical, substrate specificities, and there is no clear reason for such diversity. A possible explanation could be based on the physiological role of these isoenzymes. Functionally, they comprise a ‘‘chemical immune system’’ that must be capable of dealing with a broad spectrum of substrates, including those it has not previously encountered, and without interfering with non-toxic endogenous metabolites. A single, multifaceted enzyme would be incapable of the latter, whereas too many specific enzymes could be too costly to maintain from an energy efficiency standpoint. Perhaps the evolved optimal number of GST isoenzymes in each class is co-determined by the occurrence of xenobiotics in the distinctive environment of each species [16]. Early in mammalian development, rapid gene duplication and subsequent adoptive evolution of the replicated genes is essentially a divergent process, but may eventually have resulted in multiple isoenzymes acquiring specificities for a given substrate by convergent evolution rather than by a common ancestry.

Traditionally, the catalytic activities of GSTs are measured with 1-chloro-2, 4-dinitrobenzene (CDNB) and cumene hydroperoxide (CuOOH) as substrates. Because they also possess selenium-independent GPx activity, GSTs are able to reduce hydroperoxides of phospholipids and free fatty acids as well as cholesterol hydroperoxides [17,18,19]. In this regard, GST’s can impact the regulation of certain electrophilic intermediates indirectly controlling critical regulatory pathways. For example, 4-hydroxy-2-trans-nonenal (4-HNE), is a potentially toxic stable end product of lipid peroxidation, a common denominator in stress-mediated signaling and a pro-apoptotic second messenger that alters cell cycle signaling pathways in a concentration-dependent manner [20,21]. Steady-state intracellular levels of 4-HNE are maintained by the balance between its production, due to lipid peroxidation, and its removal via different pathways. GSTs are a major determinant of the intracellular concentration of 4-HNE as they catalyze the conjugation of GSH with 4-HNE [22]. In particular, the GSTA4-4 isoform possesses a higher affinity for 4-HNE [23] than other xenobiotics, implying a critical role in regulating 4-HNE homeostasis. The GS-HNE adduct formed is then transported out of the cell in an ATP-dependent manner, similar to the system that exports other GSH conjugates [24,25] (Figure 1).

It is generally accepted that the conjugation of GSH with xenobiotics almost always results in the formation of less reactive metabolites that are more readily excreted. However, in some cases, GSH conjugates can be more reactive than their parental compounds. Examples include short-chain alkyl halides bearing two functional groups. The conjugation of GSH with dichloromethane results in the formation of highly unstable s-chloromethylglutathione adducts, containing an electrophilic center capable of modifying DNA [26,27], with subsequent toxic effects. Cisplatin, a commonly used anticancer agent, leads to nephrotoxicity [28]. This occurs when GSH conjugates of platinum are metabolized in the proximal tubule cells of kidneys [29,30]. It was subsequently shown that this occurs in a GSTpi-dependent manner using both genetic and pharmacological inhibition in vivo [31,32]. Cisplatin-induced nephrotoxicity could be diminished using GSH mimetics [33] (Figure 2).

The capacity of cells to maintain cellular redox homeostasis during oxidative stress resides in their ability to induce a battery of protective enzymes critical to mounting a cellular defense against ROS/RNS or toxic electrophiles. In this context, the Nrf2/Keap1 transcription complex in animals is the primary finely tuned system that regulates the expression of many oxidative stress-related genes, including antioxidant and phase II detoxification enzymes. This is achieved via interactions with antioxidant response elements (ARE) in their promoter regions [34,35]. Under unstressed conditions, Nrf2 binds to Keap1 in the cytosol where it is instantly ubiquitinated and destined for proteasomal degradation. However, under conditions of elevated oxidative stress, Keap1 becomes oxidized, and this inhibits its binding to Nrf2. The activation and translocation of Nrf2 into the nucleus is facilitated by various cellular protein kinases [36,37]. Once inside the nucleus, Nrf2, along with other small Maf proteins, binds to the ARE of target genes and induces their expression. In this way, the induction of antioxidant and phase II detoxification enzymes contributes to cytoprotection against chemical insults that cause direct or indirect oxidative stress.

## 3. Role in Protein S-Glutathionylation

Protein post-translational modifications (PTMs) complement functional proteomics by regulating enzymatic activities, stability, localization, and their interactions with other cellular proteins [38]. Because of the valence flexibility of sulfur, the redox reactions of cysteine thiols can be exceptionally dynamic. Oxidative stress can preferentially react and oxidize protein thiolates (RS^−^). Protein S-glutathionylation is a redox-sensitive, reversible PTM that adds GSH to a cysteine residue in a acceptor protein [39,40]. Cysteines on the surfaces of globular proteins are generally readily accessible to GSH and GSSG and can undergo spontaneous S-glutathionylation [41], which can be altered by antioxidant enzyme systems such as thioredoxin (Trx) [42], glutaredoxin (Grx) [43], or sulfiredoxin (Srx) [44]. Grx isoenzymes have been shown to metabolize both glutathionylation and deglutathionylation reactions and are controlled by the overall redox state of the GSH pool, where a more oxidized (GSSG) environment favors glutathionylation while restoration of reduced (GSH) levels steers deglutathionylation [45].

Of relevance, a prerequisite for a cysteine residue modification is its accessibility by the solvent and reactivity influenced by adjacent amino acids. The estimated pKa of the cysteine thiol under physiological pH ranges from 8.0 to 8.7, contributing to low reactive potential. GSTs can effectively lower this pKa of the cysteine thiol, creating a more reactive nucleophilic thiolate anion [46]. Several GST isoenzymes have been reported to facilitate S-glutathionylation reactions. In particular, GSTP1-1 has been shown to catalyze glutathionylation of numerous cellular proteins, particularly under oxidative stress conditions both in vitro [9,47,48] and in vivo [10,49].

Peroxiredoxins (Prdxs) are important thiol-dependent peroxidase enzymes that are ubiquitously expressed and are known targets for GSTP1-1 mediated reversible S-glutathionylation. They perform their antioxidant functions by using intracellular thiols to catalyze the reduction of H_2_O_2,_ and other alkyl hydroperoxides. There are two major sub-classes of peroxiredoxins, 1-cys Prdx (commonly known as Prdx VI) and 2-cys Prdx. It has been shown that the catalytically active cysteine residue of Prdx VI undergoes oxidation and results in the loss of peroxidase activity. The heterodimerization of Prdx VI with GSTP1-1 facilitates the S-glutathionylation of the previously oxidized catalytic cysteine residue and restores the enzyme’s peroxidase activity [50]. Interestingly, studies suggest that polymorphic variants of GSTP1-1 may differentially mediate the activation of Prdx VI and hence influence an individual’s response to oxidant stress. For example, GSTP1-1A, is the most abundant variant of GSTP1-1 and shows a higher affinity for Prdx VI than those of GSTP1-1B or 1D variants. Furthermore, the transient transfection of GSTP1-1A in MCF-7 breast cancer cells exhibited higher peroxidase activity than that of the GSTP1-1B variant. The variations in catalytic activity between different polymorphic isoforms could be attributed to the relative distance between oxidized cysteine of Prdx VI and the activated GSH bound to the GSTP1-1 molecule [51]. Such information might imply that polymorphisms in the human population can regulate response to oxidative stress and influence factors in responding to such stress.

S-glutathionylation reactions have also been shown to influence the functions of many endoplasmic reticulum (ER) resident proteins involved in regulating the unfolded protein response (UPR). Protein disulfide isomerase (PDI) is a multifaceted ER resident protein that plays a pivotal role in cellular protein folding through its chaperone and isomerase activities. GSTP1-1 has been shown to S-glutathionylate PDI in cells exposed to oxidative or nitrosative stress, including cigarette smoke [52,53]. GSTP1-1-mediated S-glutathionylation of PDI results in impaired isomerase activity and is potentially an upstream signaling event in UPR that could influence the functionality of other client proteins with enormous impact on cellular proteostasis [52]. Although, GSTP1-1 is primarily considered a cytosolic protein, its sub-cellular distribution in the nucleus [54], mitochondria [55], and ER [49] has been reported.

Non-enzymatic S-glutathionylation reactions depend upon the GSH/GSSG ratio within the cell and occur through thiol-disulfide exchange reactions between GSSG and the protein cysteinyl residues, or by the reaction of GSH with an oxidized thiol derivative such as S-nitrosyl (-SNO), thiyl radical (-S•), or sulfenic acid (-SOH) (Figure 3) [56].

The importance of S-glutathionylation as a post-translational modification in modulating cellular processes is underscored by the ubiquity of these reactions in mammalian cells. Indeed, reversible S-glutathionylation reactions have been shown to control an array of cellular processes that include calcium homeostasis, cell signaling, chemotaxis, immune cell function, energy metabolism, and glycolysis [57]. Moreover, deregulated glutathionylation reactions have been shown to be involved in pathogenesis of many human diseases, including cardiovascular, neurological, and metabolic disorders, cataracts, and impaired embryonic development [50]. Collectively, these reactions are critical to the oversight of cellular protection against environmental insults of many different types.

## 4. Role in Signaling

Cells are constantly exposed to internal or external stressors that trigger signaling cascades and result in the activation of numerous biological processes such as cell stress response, differentiation, proliferation, and apoptosis. Control of these pathways is complex and is regulated by upstream activation of members of the mitogen-activated protein kinase (MAPK) family. Jun-terminal kinases (JNKs) are a sub-class of MAPK kinases, initially identified as stress-activated protein kinases in mouse liver treated with cycloheximide to induce inflammation and apoptosis [58]. JNKs are transiently activated by several stress stimuli including ROS/RNS, UV irradiation, heat or osmotic shock, or inflammatory cytokines [59]. JNK activation can result in subsequent phosphorylation of multiple nuclear substrates that can include transcription factor c-Jun, activating transcription factor 2 (ATF2), p53, and others, and further stimulate downstream targets and contribute to stress response through alterations in the cell cycle, DNA damage repair, and/or programmed cell death [59].

GSTs exhibit significant ligand-binding properties, and several GST isoenzymes have been shown to interact with stress kinases during regulation of cell signaling pathways responsible for stress response, cell proliferation, and apoptosis. Acting in a non-enzymatic chaperone role, GSTP1-1 negatively regulates signaling by sequestering the JNK kinase in a complex, preventing its capacity to act upon its downstream effectors. In unstressed cells, the basal activity of JNK is necessarily maintained at low levels by sequestration within the protein complex that includes at least GSTP1-1 and JNK [9]. However, under oxidative or chemical stress conditions, GSTP1-1 dissociates from the complex and accumulates in oligomeric structures, resulting in the release and activation of JNK for subsequent phosphorylation of downstream targets regulating cell proliferation and apoptosis [60]. Furthermore, GSTP1-1 has been documented to interact with, and inhibit, tumor necrosis factor (TNF) receptor-associated factor 2 (TRAF2), an upstream regulator of JNK, hence modulating the MAPK signaling cascade at multiple levels. The inhibitory effects of GSTP1-1 on TRAF2 were shown in human cervical cancer cells, where overexpression of GSTP1-1 suppressed TRAF2-induced activation of both JNK and p38. Additionally, GSTP1-1 has been shown to weaken the effects of TRAF2 on apoptosis signal-regulating kinase-1 (ASK1) and inhibit TRAF2-ASK1-induced apoptosis by suppressing the interaction between these two proteins. In comparison, reducing GSTP1-1 levels triggers TRAF2-ASK1 association and results in the activation of both ASK1 and JNK [61]. Biochemical analysis of the complex revealed that GSTP1-1 interacts with TRAF2 through both G and H sites. Even though the engagement of GSTP1-1 with TRAF2 is dimeric in form, only one monomer is involved in binding with TRAF2 and therefore the other monomer may still perform catalytic functions [62].

Furthermore, GSTP1-1 has been implicated in modulating the transcription factor nuclear factor kappa B (NF-κB) [63], which promotes the activation of pro-inflammatory signaling cascades [64,65]. An in vitro study using unstimulated mouse lung alveolar epithelial cells showed a constitutive association between GSTP1-1 and IκBα, and resulted in the inhibition of NF-κB, potentially by preventing the phosphorylation and ubiquitination of IκBα. However, LPS stimulation led to a rapid decrease in GSTP1-1/IκBα association and increased interaction between GSTP1-1 and IKKβ, along with increased IKKβ-SSG levels. These results were supported by decreased S-glutathionylation of IKKβ after siRNA-mediated knockdown of GSTP1-1 in LPS exposed cells. GSTP1-1 ablation also promoted NF-κB nuclear translocation, transcriptional activity, and pro-inflammatory cytokine production, suggesting a potential inhibitory activity of GSTP1-1 on IKKβ [63]. Similar results to that of GSTP1-1 knockdown were observed by using isotype-selective GSTP1-1 inhibitor TLK 117, which also enhanced NF-κB transcriptional activity and pro-inflammatory cytokine production in LPS-treated cells, indicating that GSTP1-1 catalytic activity is essential in repressing NF-κB activation. Such results suggest that S-glutathionylation of IKK proteins may represent a model through which GSTP1-1 can attenuate NF-κB [63]. The possible mechanistic model anticipates that, in the absence of a stimulus, GSTP1-1 averts IκBα degradation and GSTP1-1-mediated S-glutathionylation shuts down IKK activity, providing a versatile mechanism by which GSTP1-1 represses NF-κB activation.

Notably, the direct interaction of GSTs with MAP Kinases is not limited to GSTP1-1. Other GST isoenzymes have also been shown to be involved in regulating MAPK signaling pathways. For example, GSTM1-1 can bind to, and inhibit, the activity of ASK1. Under stress conditions, the GSTM1-1–ASK1 complex dissociates, causing the oligomerization of GSTM1-1 and the activation of ASK1, which subsequently activates JNK and P38 pathways, leading to apoptosis [66]. Elevated expression of GSTM1-1 has been associated with an impaired clinical response to therapies in a number of different types of cancers.

GSTA1-1 can also bind to and suppress activation of JNK signaling by pro-inflammatory cytokines or oxidative stress, implying a protective role in JNK-mediated apoptosis [67]. More recently, it has been documented that GSTA1-1 negatively regulates the mTOR signaling pathway and the over-expression of the isoenzyme in hepatocellular carcinoma cells showed enhanced AMPK activity and subsequent inhibition of the mTOR pathway. Moreover, cancer patients with high expression of GSTA1-1 in their tumors had better prognoses, and the isoenzyme may serve a protective role against hepatocellular carcinoma by suppressing the AMPK/mTOR-signaling pathway [68]. Additionally, GSTO1-1 has been shown to modulate Akt and MEK1/2 kinase pathways in human neuroblastoma SH-SY5Y cells, where the catalytic activities of the isoenzyme was shown to suppress the activation of these two kinases (either directly or in a complex) to maintain basal levels [69].

## 5. GST Polymorphism and SARS-CoV-2 (COVID-19) Disease Susceptibility

COVID-19 is a highly infectious disease caused by severe acute respiratory syndrome coronavirus 2 (SARS-CoV-2). The disease emerged in late 2019 and was declared a global pandemic in March 2020 by World Health Organization. Severe illness and death rates were more prevalent in elderly people and in those having underlying health conditions. However, the majority of the individuals infected with COVID-19 showed mild symptoms and did not require hospitalization. The severity of the infection was driven by the host’s response to the disease, triggering a cascade of inflammatory responses and respiratory dysfunction [70]. Individuals with COVID-19 infections have been reported to have markedly higher levels of inflammatory cytokines that trigger a pro-inflammatory response and cause tissue damage, thus contributing to the severity of the disease [71,72]. There is mounting evidence to support the concept that inflammatory disease progression is associated with increased ROS production and resultant oxidative stress. Imbalanced redox homeostasis linked to COVID-19 [73] may contribute to inter-individual differences in patient clinical manifestations, influenced by genetic variation in antioxidant enzyme systems. GST polymorphisms are common in humans and range from frequencies as high as 20% to 60 % in some populations, including the null genotypes of GST Mu and Theta class (GSTM1^−/−^ and GSTT1^−/−^, respectively), whereby individuals lack a catalytically active enzyme [74]. In a study comprising 269 RT-PCR-confirmed COVID-19 patients (with both mild and severe conditions), Abbas et al. [75] reported the relationship between GSTM1 and/or GSTT1 genotypes with COVID-19 vulnerability and its outcome in Northern Indian populations. The results indicated that the frequencies of GSTM1^−/−^, GSTT1^−/−^ and GSTM1^−/−^/GSTT1^−/−^ were more pronounced in patients with severe COVID-19 symptoms than those with mild. Overall, patients with GSTT1^−/−^ genotypes had higher mortality rates than those with GSTT1^+/+^. In another study, Saadat [76] suggested that individuals with the GSTT1^−/−^ genotype showed a positive association with COVID-19 mortality, but no correlation with COVID-19 prevalence. However, individuals with a low frequency of the GSTT1-null genotype exhibited higher numbers of COVID-19 deaths in East Asian countries.

Tatjana et al. [77] recently reported in Serbian populations that individuals carrying GSTO1*AA and GSTO2*GG polymorphic variants had a significantly increased risk of COVID-19 development as compared to the wild-type genotype. Vesna et al. [78] studied the distribution of GST genotypes among COVID-19 patients of both genders with common COVID-19 co-morbidities such as hypertension, diabetes, and obesity. The results indicated a significant association between GSTP1 and GSTM3 polymorphisms and COVID-19 susceptibility and clinical manifestation. Individuals carrying the GSTP1* (Ile105Val rs1695) or GSTP1* (Ala114Val rs1138272) variants were less prone to develop COVID-19 as compared to the GSTP1 wild type genotypes. Similarly, individuals with GSTM3*AC (rs1332018) variants showed lower odds of developing COVID-19 compared to the wild type GSTM3. The combined GSTP1* and GSTM3* polymorphisms showed a cumulative risk regarding COVID-19 prevalence and severity. These results have shed some light on the involvement of genetic susceptibility in COVID-19 development and further pointed out the multifaceted role of GSTP1-1 as being the dominant GST class in lungs.

Extensive research efforts have documented the relevance of GST polymorphisms in governing cancer susceptibility [65,79,80], and with disease progression or response in various communicable or non-communicable lung diseases [65,81]. In particular, the homozygous GSTP1* Val allele was found to be associated with a reduced risk of asthma and improved lung function [82]. Ding et al. [83] reported the association of GSTT1 and/or GSTM1-null genotypes with an increased risk of developing pulmonary fibrosis in patients with chronic obstructive pulmonary disease (COPD), one of the more important COVID-19 complications characterizing long-term respiratory problems.

Taken together, the polymorphic studies reviewed in the current article revealed that GSTT1 and GSTM1-null genotypes show differential behavior against COVID-19 mortality. Individuals with a lower frequency of GSTT1-null genotype showed higher COVID-19 mortality rates, while GSTM1-null genotype increased the odds of severe disease outcome. The carriers of GSTM3* (rs1332018) and GSTP1* (rs1695) heterozygotes and GSTP1* (rs1138272) Val allele showed a reduced risk of developing COVID-19 as compared to the wild-type carriers. The cumulative effect of GST gene polymorphisms on COVID-19 disease development shows that the individuals with a higher number of risk-associated genotypes had a higher risk of developing COVID-19.

## 6. GST Inhibitors and Their Therapeutic Importance

GSTs emerged as viable therapeutic targets because specific GST isoenzymes have been shown to be over-expressed in numerous tumors. Involvement in the pathogenesis of other diseases including allergic asthma, multiple sclerosis, and neurodegenerative diseases has also been considered [84,85,86,87]. Over the years, attempts have been made to develop specific and potent inhibitors of GSTs with the goal of diminishing tumor growth and enhancement of the cytotoxic effects of existing chemotherapeutic agents [88,89,90,91]. The first clinical study was carried out on ethacrynic acid, an approved diuretic drug that nonspecifically inhibits GSTα, GSTµ, and GSTπ isoforms [92]. Although ethacrynic acid showed encouraging inhibitory properties in several cancers, its strong diuretic properties and lack of isozyme specificity made it less favorable for its clinical use as a modulator of anticancer drugs. A further disadvantage was its propensity for enzymatic cleavage as the γ-glu-cys peptide bond in the GSH conjugates is sensitive to γ-glutamyl transpeptidase enzymes. Thus, attempts have been made to develop GSH analogues/prodrugs with improved stability and clinical properties. Newer ethacrynic acid derivatives have been developed and tested for their antiproliferative properties in vitro [93]. Ethacraplatin is a platinum (IV)-based prodrug designed to overcome GST mediated cisplatin resistance. The cisplatin molecule is incorporated between two ethacrynate ligand moieties that inhibit the enzymatic activity of GSTP1-1 and liberates the cisplatin molecule as a consequence of binding, which in turn results in the increased localized diffusion of Pt ions and reversion of platinum drug resistance [94].

NBDHEX (6-(7-nitro-2,1,3-benzoxadiazol-4-ylthio) hexanol) is another inhibitor of GSTP1-1 and other GSTs that has shown anti-proliferative activities in various cancer cells [95,96]. NBDHEX acts as a mechanism-based inhibitor of GSTP1-1, where it is first recognized as a substrate and makes a spontaneous intermediate σ- complex with GSH, which binds very tightly to the enzyme and results in the loss of both the GSH-conjugating activity as well as the ability to form complexes with partner proteins JNK1 and TRAF2 [62,95,97]. NBDHEX was tested on several human osteosarcoma cell lines resistant to cisplatin, doxorubicin or methotrexate and proved to be active against the majority of drug-resistant cell lines [98].

GSH is the most abundant low molecular weight thiol in the cell and a number of GST inhibitors have been designed on the structural basis of the GSH moiety. Ezatiostat hydrochloride (Telintra, TLK199) is a peptidomimetic analogue of GSH and a well characterized inhibitor of GSTP1-1 [15]. Studies conducted on mouse fibroblasts, showed that after intracellular de-esterification to TLK117, it binds to and inhibits GSTP1-1 and can disrupt the binding of GSTP1-1 to JNK. This results in the activation and restoration of JNK and MAPK pathways, which subsequently promotes MAPK-mediated cellular proliferation and differentiation pathways [99]. In addition to GSTP1-1 inhibition, TLK199 has been shown to effectively inhibit multidrug resistance-associated protein 1 MRP1 (encoded by ABCC1 gene), which plays an essential role in multidrug resistance by its ability to effectively export an array of chemotherapeutic and other drugs out of the cell [100]. Studies conducted on MRP1-transfected NIH3T3 mouse fibroblasts with minute GSTP1-1 levels, TLK199 showed significant inhibition of ATP-dependent efflux and resulted in the enhanced retention and subsequent reversal of numerous resistant phenotypes including doxorubicin, daunorubicin, etoposide, mitoxantrone and vincristine [101]. Additionally, in phase I/II clinical studies, TLK199 was tested in patients with myelodysplastic syndrome (MDS), a diverse group of bone marrow stem cell disorders largely affecting individuals with median ages of 65–70 years at diagnosis. The results of ezatiostat treatment in patients with lower to mild risk MDS were encouraging in patients with trilineage cytopenias. Hematologic Improvement-Erythroid (HI-E) was observed in 11 out of 38 patients (29% cases), HI-Neutrophil (HI-N) was reported in 11 out of 26 patients (42% cases), while HI-Platelet (HI-P) was observed in 12 out of 24 patients (50% cases). HI-E was also reported in a few patients with red blood cells (RBC) transfusion dependency with no prior record of receiving any therapy with hypomethylating agents [15,102]. Overall, the available clinical data showed encouraging results for ezatiostat in MDS patients with favorable tolerability and hematopoietic-promoting activities, indicating the worthiness of the drug for further evaluation in randomized phase II/III clinical trials.

Idiopathic pulmonary fibrosis (IPF) is a chronic and progressive lung fibrotic disease that results in the thickening of alveolar walls and diminished lung function [103]. The pathogenic mechanism of the disease is not yet completely understood. Hence, therapies of IPF still remain a clinical challenge. Changes in the expression levels of GSTs have been reported in pulmonary fibrosis cells, murine models, and in IPF patients [15,81,104]. This suggests an important role for GSTs in pulmonary fibrosis. Oropharyngeal administration of TLK117 has been shown to reduce the severity of pulmonary fibrosis. Furthermore, the synergetic effects of TLK117 with pirfenidone showed better therapeutic outcomes in pulmonary fibrosis mice models than by using pirfenidone alone [104].

To date, clinical experiences with GST inhibitors have been limited. Ethacrynic acid in both animals and humans caused the expected diuretic effects and the Phase II clinical trial was stopped because of severe fluid and electrolyte imbalance [105]. This had less to do with the GST-inhibitory effect but was more related to the thiol-mediated diuresis’ impact on the kidney [106,107]. In Phase I/II clinical trials, Telintra was dose-limited by the unusual toxicity of patients displaying unpleasant sulfurous odors, primarily because of metabolism of the peptidomimetic [15]. While there are now reports of the involvement of GST isozymes in myeloproliferation, none of the GST inhibitors so far developed appear to be restricted in their pharmacological effects by myelotoxicities. In summary, the rodent model and early clinical experiences with direct GST-inhibitory drugs would suggest that neither acute nor chronic treatments are accompanied by serious dose-limiting toxicities.

Canfosfamide (TLK 286, TELCYTA) is a glutathione analogue prodrug that is preferentially activated by GSTP1-1 into a vinyl sulfone derivative of the GSH backbone and an alkylating metabolite phosphorodiamidate that spontaneously forms aziridinium ring structures [108,109,110]. The rationale behind designing this prodrug was to address the issue of anticancer drug resistance due to overexpression of GSTP1-1 in many tumors and to limit the off-target adverse effects. In vitro studies showed that TLK286 had higher antiproliferative activity in cells overexpressing GSTP1-1. In vivo studies using xenograft models in nude mice showed that tumor growth inhibition or regression was positively corelated with GSTP1-1 expression levels in response to TLK286 treatment, and a mild bone marrow toxicity was observed as a side effect [111]. These promising results led to several clinical studies where the prodrug was demonstrated to be active and safe, as a single agent or in combination regimens with other established drugs, including anthracyclines, platinums, and taxanes. The clinical efficacy of the prodrug in phase II and III clinical trials was observed in both relapsed patients with ovarian and non-small cell lung cancers, and in the first-line treatment setting in non-small cell lung cancer patients [112,113].

Advances in structural studies have been instrumental in designing isotype-specific inhibitors since different GST isoforms have unique glutathione binding sites [114]. The efforts to use glutathione as a prototype to develop G-site specific inhibitors provides an efficient platform. However, higher levels of intracellular GSH present a challenge for developing G-site-specific inhibitors [115,116]. Shishido et al. have developed GSH derivatives by introducing a sulfonyl fluoride (SF) onto the sulfhydryl group of GSH, GS-ESF, that irreversibly inhibited GSTP1-1 by a covalent bond with Tyr 108 of the isozyme [117]. However, due to the polarity of the GSH moiety, the cell permeability presented a challenge that was addressed by introducing cell-membrane-permeable benzene sulfonyl fluoride (BSF)-type covalent inhibitors [118]. The in vitro studies of these covalent inhibitors showed a prolonged inactivation of the GSTP1-1 in human non-small-cell lung adenocarcinoma cells, which served as lead compounds for the further development of potent inhibitors of GSTP1-1 in cancer.

A library of 20 dichlorotriazine probes was synthesized by Crawford et al. via tosylating 4-pent-yn-1-ol and evaluated for covalent protein labeling in Hela cell lysates [119]. Of all the compounds investigated, only one showed the potency and specificity for covalent modification of GSTP1-1 in cellular context. Mass spectrometry and mutagenesis analysis identified Y108 as the site of covalent modification of GSTP1-1 by LAS17, thus providing a unique mode of irreversible inhibition by targeting a functional tyrosine residue (Y108). Cell cultures treated with LAS17 showed impaired GST activity and reduced cell survival [120]. Daily administration of LAS17 significantly reduced breast tumor xenograft growth.

More recently Jeffery et al. [121] introduced sulfur-triazole exchange (SuTEx) chemistry as an adaptive platform for developing covalent probes with broad applications for chemical proteomics and protein ligand discovery purposes. They have previously demonstrated that modifications to the triazole leaving group can equip sulfonyl probes with enhanced chemoselectivity for tyrosines (over other nucleophilic residues). By using these probes, they identified that the tyrosine residues with enhanced nucleophilicity are more enriched in enzymatic, protein–protein interactions and nucleotide recognition domains [89]. GSTP1-1 has a reactive tyrosine Y8 in its active site and a known site for phosphorylation [122]. By using SuTEx fragments, they discovered JWB152 and JWB198 as efficient ligands of GSTP1-1 Y8. In vitro studies revealed equivalent inhibitory activities for both ligands, however, only JWB198 could ligand the Y8 site of GSTP1-1 in live cells. Proteome-wide reactivity evaluations of JWB198 were encouraging as it maintained ~70% blockade of GSTP1-1 Y8 in live DM93 cells while being substantially less reactive to the other tyrosine residues within the protein [121].

## 7. Conclusions

The various functional properties of GSTs have drawn a lot of attention from researchers all over the world for decades. Initially, the enzymes were best known for their catalytic functions in the detoxification process of endogenously produced and or xenobiotic electrophiles. However, advances in the field have brought research to focus on additional biologically important roles ascribed to this versatile enzyme family including cell signaling and post-translational modifications, as well as conferring resistance to chemotherapy, since many GST isoforms have been shown to be overexpressed in numerous cancers. In this context, it is not surprising that a large number of GST inhibitors and prodrugs have been designed and tested for their therapeutic applications. Some of these inhibitors have entered into phase II/III clinical trials, and in the future we may welcome the approval of GST inhibitors/prodrugs as therapies for patients. Moreover, the data on the effects of GST genetic polymorphisms on COVID-19 disease severity and clinical manifestations is inconclusive, but it may add to our understanding of the risk factors that contribute to the severity of the disease and may potentially be useful for better selecting targeted pharmacological strategies for individual COVID-19 patient needs.

## 8. Perspectives

There has been much progress since the discovery and early descriptions of GST isozymes in the 1960s. Their role in detoxification have been described in detail and the isozyme family provides a broad-ranging capacity to conjugate small electrophilic chemicals with GSH. While the early years focused attention on the catalytic properties of isozymes and their organ distribution, subsequent publications have served to exemplify their ligand binding properties, influence on cell signaling, thioltransferase activities and their general involvement in human pathologies. Pharmacology, particularly in cancer, has produced a variety of small molecular drugs that either inhibit GST pathways or serve as prodrugs that might be activated by GST. Since cancer is characterized by aberrant energy production and signaling pathways, as well as maintaining a redox homeostasis distinct from normal cells, there has been much interest in how GST expression patterns may differ in malignant disease and how they might be targeted. Many types of cancer and drug-resistant tumor cells express extremely high levels of GST, particularly GSTP, and oncology studies continue to extend the role of this isozyme in regulating signaling pathway cascades. Some of these studies consider the involvement of GST in regulating energy production and may yield information pertinent to how cancer cells use glycolysis versus the pentose phosphate pathway, the so-called Warburg effect. Moreover, post-translational modification of the proteome extends structure/function properties of many proteins, and the involvement of, for example, GSTP1-1 and GST omega in S-glutathionylation and deglutathionylation reactions. There is a continually expanding literature documenting the substantial corpus of proteins that are subject to this cysteine modification. In brief, despite the nearly 60 years of published studies documenting GST, there still remains a significant amount to be learned about this ubiquitous and adaptable family of enzymes.

## Figures and Tables

**Figure 1 biomolecules-13-00688-f001:**
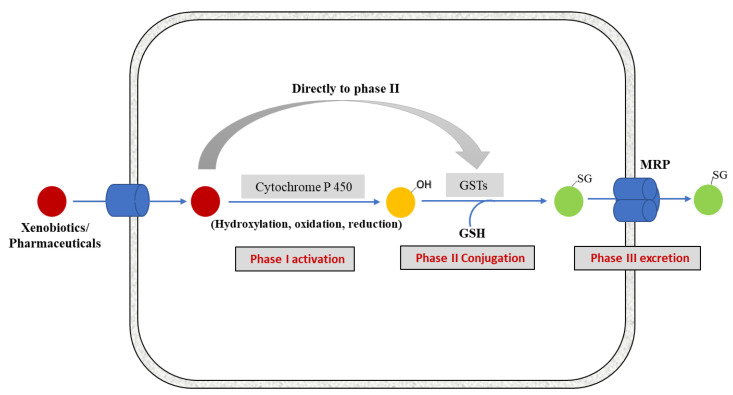
Overview of xenobiotic biotransformation pathway. Once inside the cell, the toxic molecules are targeted by different enzymes of detoxification system. Lipophilic molecules are metabolized by Phase I enzymes, i.e., Cytochrome P450s. The activated xenobiotics are subsequently conjugated with GSH by phase II detoxification enzyme GSTs and are finally exported out of the cell in phase III by trans-membrane multidrug resistance-associated proteins (MRPs) from the C family of ABC transporters [1]. Some compounds (polar or hydrophilic in nature) may enter in Phase II metabolism directly.

**Figure 2 biomolecules-13-00688-f002:**
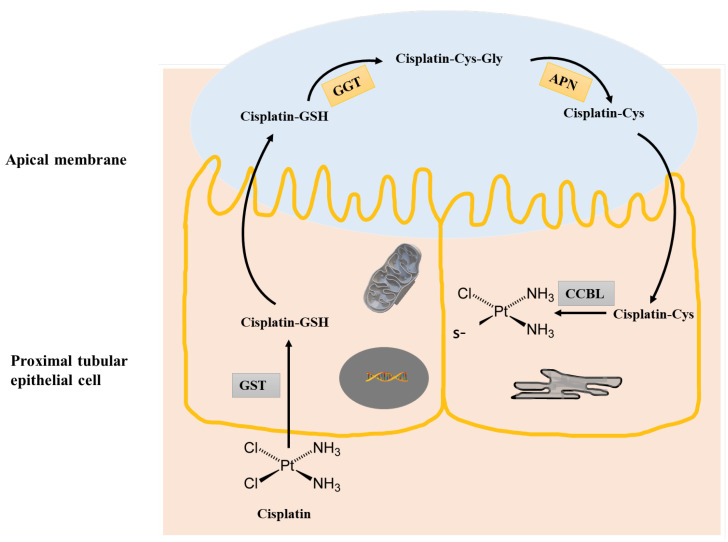
Bioactivation of cisplatin to a nephrotoxin. In the first step, Cisplatin can be conjugated with GSH by GST. The Cisplatin–GSH conjugate then passes through to the tubule lumen by MRP2 effluxes, where it is cleaved to a cysteinyl–glycine conjugate by g-glutamyl transpeptidase (GGT) followed by further cleavage to Cisplatin–Cysteine by aminopeptidase (APN). The Cisplatin–Cys conjugate is then re absorbed by the proximal tubule, where Cisplatin–CYS is further metabolized by a pyridoxal 5′-phosphate-dependent enzyme, cysteine S-conjugate beta-lyase (CCBL) to form a reactive thiol that can bind proteins and contribute to toxicity.

**Figure 3 biomolecules-13-00688-f003:**
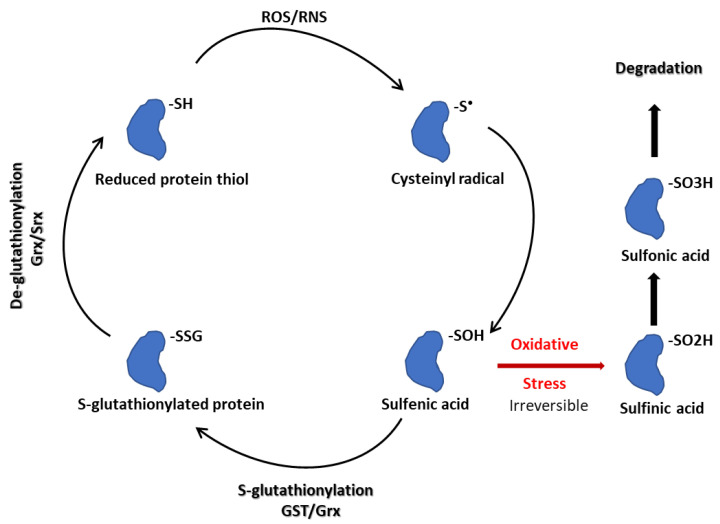
S-glutathionylation schematic.

## Data Availability

Not applicable.

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
