# Peer review of "The Multifaceted Role of Glutathione S-Transferases in Health and Disease"

_biomolecules, 2023, doi:10.3390/biom13040688_

Round 1

Reviewer 1 Report

The paper from Mazari and colleagues is a good, straightforward review of the glutathione transferase field. The review includes recent work on potential links to COVID19, which will be of topical interest to readers.

I have some comments for the authors to address:

I would like to see more discussion on tissue distribution/specificity of the transferases. Although the authors claim they are expressed in every cell, there are some notable differences in expression which reflect functional importance e.g. GST1 is only weakly expressed in adipose tissue and muscle.

I would also like to see some discussion about intracellular targeting of transferases. In the section on glutathionylation of PDI, for example, how does GSTP1 access PDI, given PDI is mainly resident in the ER, whereas GSTP1 is cytosolic?

Please could the authors comment on the long term risk of complications, if GST inhibitors are to be used to treat cancer and COVID19.

Minor points:

On Pg 8 – it is not clear what is meant by “current forum article”.

There are a few spelling/typographical issues e.g in the abstract: cytosolic glutathione S-transferases family should be cytosolic glutathione S-transferase family; in the introduction detoxication should be detoxification; font formatting issue on Pg 3 “this occurs when GSH …’.; Figure 2 should be (Figure 2).

Author Response

Dear Reviewer, 

Comments and Suggestions for Authors

The paper from Mazari and colleagues is a good, straightforward review of the glutathione transferase field. The review includes recent work on potential links to COVID19, which will be of topical interest to readers.

I have some comments for the authors to address:

I would like to see more discussion on tissue distribution/specificity of the transferases. Although the authors claim they are expressed in every cell, there are some notable differences in expression which reflect functional importance e.g. GST1 is only weakly expressed in adipose tissue and muscle.

We have now added new material on page 2.

I would also like to see some discussion about intracellular targeting of transferases. In the section on glutathionylation of PDI, for example, how does GSTP1 access PDI, given PDI is mainly resident in the ER, whereas GSTP1 is cytosolic?

Though, GSTP1-1 is primarily considered a cytosolic protein, sub-cellular distribution in nuclear [55], mitochondria [56] and ER [50] has been reported.

Page 5, 5th paragraph

Please could the authors comment on the long-term risk of complications, if GST inhibitors are to be used to treat cancer and COVID19.

We now add a section outlining those issues that may impact the therapeutic index of GST inhibitors. See page 10.

Minor points:

On Pg 8 – it is not clear what is meant by “current forum article”.

Corrected to “current article”

There are a few spelling/typographical issues e.g in the abstract: cytosolic glutathione S-transferases family should be cytosolic glutathione S-transferase family; in the introduction detoxication should be detoxification; font formatting issue on Pg 3 “this occurs when GSH …’.; Figure 2 should be (Figure 2).

Corrected to suggested by the reviewer.

Reviewer 2 Report

This review summarizes the role and function of GST in Phase II  metabolism. Partucular focus is given on the involvement of GSTs on S-glutathionylation and on the effects of GST genetic polymorphisms on COVID-19 disease. The work is interesting, especially the part on COVID-19 disease. However, there are a few points which need to be addressed by the authors:

1) The connection of GSTs with the NF-κB inflammatory signaling pathway is not discussed are explained. The authors should consider adding relevant information.

2) The paragraph “GST inhibitors and their therapeutic importance” is not described with sufficiently detail and does not covered adequately. Since Drug design efforts is an important area of GST’s research the authors need to put some more effort to cover this area with resent progress and more recent literature.

Author Response

Dear Reviewer, 

Comments and Suggestions for Authors

This review summarizes the role and function of GST in Phase II metabolism. Particular focus is given on the involvement of GSTs on S-glutathionylation and on the effects of GST genetic polymorphisms on COVID-19 disease. The work is interesting, especially the part on COVID-19 disease. However, there are a few points which need to be addressed by the authors:

  • The connection of GSTs with the NF-κB inflammatory signaling pathway is not discussed are explained. The authors should consider adding relevant information.

We agreed and included the relevant information on page 5, 2nd paragraph. 

  • The paragraph “GST inhibitors and their therapeutic importance” is not described with sufficiently detail and does not cover adequately. Since Drug design efforts is an important area of GST’s research the authors need to put some more effort to cover this area with resent progress and more recent literature.

Considering article length limitations, we were reluctant, but have now included some recently designed inhibitors. See page 11 paragraph 1 and 2.

Reviewer 3 Report

This review summarizes the recent findings for the diverse role of GSTs play in human health and disease.

While the review is comprehensive, I found it very descriptive . I would suggest that the authors revise the manuscript focus on making the review more critical where possible. 

In addition, this review lacks analysis and hindsight on the subject that is reviewed. A review is not just to categorize what have been done, but most importantly, to critically analyze the information, synthesize the results from the literature, and provide fresh and insightful argument and/or discussion on the published results. 

The text is very fluid and well written.

Author Response

Dear Reviewer, 

Comments and Suggestions for Authors

This review summarizes the recent findings for the diverse role of GSTs play in human health and disease.

While the review is comprehensive, I found it very descriptive. I would suggest that the authors revise the manuscript focus on making the review more critical where possible. In addition, this review lacks analysis and hindsight on the subject that is reviewed. A review is not just to categorize what have been done, but most importantly, to critically analyze the information, synthesize the results from the literature, and provide fresh and insightful argument and/or discussion on the published results.

We have reworked some of the narrative to provide a more critical interpretation of the field and provided an overall perspective section.

Round 2

Reviewer 2 Report

The authors adequately responded to my comments.

Author Response

Dear Reviewer, 

Thank you for accepting our responses to your comments/suggestions. We have proofread the manuscript and made minor changes for consistency i.e detoxication to detoxification in the whole manuscript (with track changes function ON ).